# Machine Learning-Based Human Recognition Scheme Using a Doppler Radar Sensor for In-Vehicle Applications

**DOI:** 10.3390/s20216202

**Published:** 2020-10-30

**Authors:** Eugin Hyun, Young-Seok Jin, Jae-Hyun Park, Jong-Ryul Yang

**Affiliations:** 1Division of Automotive Technology, ICT Research Institute, Convergence Research Institute, DGIST, 333 Techno Jungang-daero, Hyeonpung-myeon, Dalseong-gun, Daegu 42988, Korea; ysjin@dgist.ac.kr; 2Department of Electronic Engineering, Yeungnam University, Gyeongsan, Gyeongbuk-do 38541, Korea; bravopark@ynu.ac.kr (J.-H.P.); jryang@yu.ac.kr (J.-R.Y.)

**Keywords:** passenger detection, CW radar, radar feature vector, radar machine learning

## Abstract

In this paper, we propose a Doppler spectrum-based passenger detection scheme for a CW (Continuous Wave) radar sensor in vehicle applications. First, we design two new features, referred to as an ‘extended degree of scattering points’ and a ‘different degree of scattering points’ to represent the characteristics of the non-rigid motion of a moving human in a vehicle. We also design one newly defined feature referred to as the ‘presence of vital signs’, which is related to extracting the Doppler frequency of chest movements due to breathing. Additionally, we use a BDT (Binary Decision Tree) for machine learning during the training and test steps with these three extracted features. We used a 2.45 GHz CW radar front-end module with a single receive antenna and a real-time data acquisition module. Moreover, we built a test-bed with a structure similar to that of an actual vehicle interior. With the test-bed, we measured radar signals in various scenarios. We then repeatedly assessed the classification accuracy and classification error rate using the proposed algorithm with the BDT. We found an average classification accuracy rate of 98.6% for a human with or without motion.

## 1. Introduction

The fact that children are dying in hot vehicles has recently become a major social issue. Thus, the European NCAP (New Car Evaluation Program) has recommended the installation of CPD (Child Presence Detection) technology on all new cars starting in 2020 [1]. Moreover, many countries’ safety regulators have also considered rules that could mandate CPD systems aimed to detect a child left in a vehicle. To support such a system, various sensors capable of detecting objects in vehicles or monitoring vehicle body status are required [2,3].

Another application of passenger detection is in electric vehicles. In electric vehicles, the heating and air conditioning functions depend on the efficiency of the battery [4]. If heating and cooling systems in vehicles can be automatically controlled for each seat, battery consumption can be decreased. Thus, to support these functions, technology to detect passengers in each seat is required.

One additional application for occupant recognition is in self-driving vehicles. The driving operation for an autonomous vehicle strongly depends on the presence or absence of passengers. That is, when occupants are riding, the comfort and reliability of passengers become very important issues [5]. Moreover, depending on whether the occupant is sleeping or moving, the self-driving style can differ. Thus, it is very important to assess the occupancy and status of passengers in every seat.

For the various applications described above, the performances capabilities of sensors to detect passenger are very important. The characteristics of these sensors are described below.

Among various methods, one very simple approach is to use a pressure sensor. However, when any object is placed on a seat, it is impossible to determine whether or not the object is a human. Another solution is to measure and use the distance from the object by means of an ultrasonic sensor. However, this method also cannot distinguish between different types of objects.

Recently, thermal infrared sensors have been attracting attention given their ability to check for the presence of passengers using human temperature. However, this method is highly sensitive to a person’s clothing or to the external temperature condition.

The use of a camera is also a very effective solution for these applications. Specifically, because stereo cameras and depth cameras can measure the distance to an object, they can recognize various motions of a human when applying deep learning with image features. However, camera sensors are limited due to the external lighting conditions. Another disadvantage is that the amount of computation for image processing is excessive. Moreover, the installation of a camera inside their vehicles may cause consumers to reject these vehicles due to privacy issues.

Recently, a radar sensor-based occupant detection system has attracted attention, as radar is robust to external conditions [6]. Moreover, radar can distinguish between a moving object and a stationary object, and these systems can also monitor vital signal of sleeping or non-moving humans in a vehicle [7].

To detect the motion of an object and to detect human vital signs using a radar sensor, popular types currently in use are impulse UWB (Ultra Wide Band) radar, FMCW (Frequency Modulated Continuous Wave) radar, and CW (Continuous Wave) radar.

In impulse UWB radar, because it is possible to measure a high-resolution range, we can distinguish chest movements due to breathing by measuring changes in distance values. Because high-resolution range detection is most advantageous, UWB radar is widely used for vital sign recognition [8,9]. Earlier work proposed the concept of detecting the vital signals of a passenger by mounting a UWB radar sensor in a vehicle [8]. In another approach [9], a UWB radar sensor was used to detect human vital signs for each seat, applying the features extracted from the detected range into the machine leaning approach, such as a SVM (Support Vector Machine). However, these two related works only focused on non-moving humans, and did not consider moving humans or other objects.

Although UWB radar is very popular, the Doppler component cannot be detected in order to distinguish between a stationary object and a moving object. Thus, an additional algorithm is required using measured distances. In addition, because this type transmits an impulse-shaped waveform in the time domain, high peak transmission is limited. This can result in a low SNR (Signal-to-Noise Ratio) over a certain distance.

Recently, because FMCW radar can measure both the distance and the Doppler information, FMCW radar has come to be commonly used in commercial applications. Moreover, when detecting changes in phases over several periods, the respiration period was extracted in earlier works [10,11]. In addition, one related study [10] presented a method that separated the vital signs reflected from two humans using a high-resolution algorithm, in that case the MUSIC (Multiple Signal Classifier) algorithm. However, neither method focused on only vital sign signal detection, nor were they intended for in-vehicle applications.

Despite the fact that the FMCW radar is widely used in commercial applications, a PLL (Phase Loop Lock) circuit is also required to synchronize the transmit waveform phases and to ensure linearity during the modulation step. Moreover, because FMCW radar can detect moving and stationary objects, extra algorithms to distinguish them are necessary.

Finally, the CW radar sensor is a popular radar sensor due to its very simple hardware structure. However, because CW radar can only receive Doppler signals, these sensors can only detect moving objects, and cannot detect the range. Thus, in CW radar, breathing signals can be measured by analyzing the Doppler signal generated from the chest movements [7,12,13]. One earlier study [12] presented the concept of recognizing a human remaining in a vehicle using CW radar by cancelling the background noise. In another work [13], an algorithm for detecting not only respiration, but also the heartbeat, was proposed. Because CW radar is somewhat sensitive to external noise, one study [7] proposed attaching an accelerometer to the radar sensor to record vibrations of the vehicle itself. However, these three related works also considered stationary humans when extracting human vital signs.

Although though CW radar sensors have various disadvantages, they can be easily applied in various applications as a low-cost senor compared to UWB radar and FMCW radar. Thus, in this paper, we employ the CW radar type to realize a passenger detection system with a very simple architecture.

To effectively confirm the existence of a human in a vehicle, we can recognize a human who is still, sleeping, or moving using the Doppler signal measured using the CW radar sensor. That is, we can determine whether or not the detected moving object is a human and can extract vital signals from a non-moving human on the seat.

If a human is moving on a seat, the echo signal of the human’s vital signs can be masked by the Doppler signal of the human’s motion. In such cases, it is difficult to determine whether a human is present or not in a vehicle using only the detected vital signal.

Moreover, if an inanimate object is moving on a seat or if the vehicle itself has vibration, the radar system should be able to recognize a Doppler signal. Thus, when using only the presence of the Doppler echo, it is impossible to determine whether or not a human is occupying a seat.

Thus, in this paper, we propose a human recognition concept as part of our effort to implement a proper passenger detection system, as shown in Figure 1. The wait mode transmits the received radar signal into the motion-detection mode and the vital sign detection mode, with both modes operating in parallel.

Based the results of both modes, in the decision mode, human recognition is determined, and the system reverts back to the wait mode.

For the vital sign detection mode, we design a simple vital sign detection algorithm to determine whether breathing is present or not. Thus, we extract one feature vector to indicate the presence of vital signs.

For the motion detection mode, we propose algorithms to determine whether or not a moving object in a vehicle is a human. In the proposed method, we use the human characteristics of non-rigid motion. That is, in the case of a human, because the radar signal is reflected from various components of the human’s torso, head, shoulders, arms, waist, and thighs, micro-Doppler effect appears. Thus, in paper, we initially generate a micro-Doppler image in the time-frequency domain. Next, we design two new feature vectors that suitably represent the characteristics of a moving human in a vehicle from the micro-Doppler image.

Finally, in the decision mode, we conduct machine learning using a BDT (Binary Decision Tree), which has a very simple structure, and the proposed three features to determine the presence of passengers in vehicles.

Thus, we extract three features using actual measurement data from a CW radar transceiver and verify the proposed machine learning-based human recognition scheme.

In Section 2, we present the proposed human recognition scheme with machine learning. In Section 3, we present the verification results using actual data from a 2.45 GHz CW radar front-end module and a real-time data acquisition module. Finally, we present the conclusion of our study and the suggestions for future work in the final Section.

## 2. Proposed Human Recognition Scheme in a Vehicle

### 2.1. Problem Definition of Human Detection in a Vehicle

In this paper, Figure 2 shows the information detected via radar sensor according to the increased level of motion of passengers in a vehicle. Here, the x-axis indicates the amount of movement by a human and the y-axis represents the Doppler frequency of the echo signal.

In a vehicle, when a passenger is sleeping or a still human is sitting on a seat, Doppler radar sensors can detect breathing signals from vital sign detection area in Figure 2. Accordingly, we can easily distinguish between a human and an inanimate object using the echo generated by the human’s respiration.

However, if a passenger moves with much motion on the seat, as shown in the motion detection area of Figure 2, the Doppler components issued by the body motion can mask most of the weak vital sign signals. In such cases, if an object is moving with considerable motion, we can detect the object using the Doppler echo level. However, with only the Doppler component, it is impossible to confirm whether or not a human has been detected.

Moreover, in the motion and vital sign hazard area, the detection of vital signs depends entirely on the amount of human movement. In other words, when a human is moving with relatively slight motion, motion and vital signals can appear together, whereas the vital signals can be immediately masked when a passenger is moving with a Doppler volume above a certain level.

Therefore, in both the motion detection area and the motion and vital sign hazard area, we cannot distinguish between a human and another object by the presence of vital signals or Doppler signals alone. To overcome this problem, we propose here a human recognition scheme using the characteristics of the echoed Doppler spectra of the object.

Generally, in signals reflected from a walking or running human, sidebands appear around the Doppler frequency due to the non-rigid motion. That is, various Doppler echoes can be extracted as reflected scatters of human components, such as the body, arms, and feet. Moreover, the distribution of Doppler scattering points received from a human can vary greatly during the measurement time [14,15]. In one earlier study [14], based on the FMCW radar, we proposed a concept to distinguish between pedestrians and vehicles on a road by detecting the range and velocity and analyzing the pattern of the Doppler spectrum. In addition, in another study [15], we proposed a classification algorithm for humans and vehicles that extracted feature vectors from the received FMCW radar signal and applied them to machine learning.

From these hints, in vehicle applications, although the received power is weak and there are fewer scattering points compared to the case of a walking human, the received signal has multiple reflection points echoed from the head, torso, waist, arms, pelvis, and other parts of the passenger’s body. Moreover, because the human in the vehicle cannot move constantly, the Doppler spectrum will vary more over time.

Thus, we can find certain patterns in the micro-Doppler image of a moving passenger. In this paper, we distinguish between humans and other objects by using feature vectors extracted from this pattern, together with the vital signal extracted through additional signal processing, and applying the features to machine learning.

### 2.2. Concept of Proposed Human Recognition

Figure 3a presents the proposed concept of recognizing a passenger in a vehicle based on the Doppler spectra and vital signs of the passenger. In Figure 3b, we illustrate an example of the scattering points reflected from the passenger and from a box. In this case, we cannot know which aspects of the human’s various components are reflected. Moreover, we cannot know the extent to which individual parts of body contribute to the reflection.

However, as explained above, if a human is moving, the Doppler spectrum reflected by the human is expected to spread and change over time due non-rigid motion, as shown in Figure 3b. On the other hand, echoes from a non-human object such as a moving box are expected to have a sharp type of Doppler spectrum, as shown in Figure 3c. These characteristics are the motivation behind the algorithm proposed in this paper.

The proposed human recognition scheme is divided into two parallel signal processing parts: the micro-Doppler-based motion feature extraction part and the Doppler frequency-based vital sign feature extraction part.

The I and Q signals reflected from the object are sampled through an ADC (Analog Digital Converter). In this paper, we set the ADC sampling rate to 1 kHz, which is a high enough value to digitalize the received signal form the CW radar system.

The corresponding digitized data are inputted into two signal processing parts. In the micro-Doppler-based motion feature extraction part, we can extract two feature vectors (*x1* and *x2*) that can determine whether or not human motion appears. Details are presented below in Figure 4 and Figure 5.

In the Doppler frequency-based vital sign feature extraction part, the third vector *x3* is extracted to determine whether the vital signs exist. Figure 6 shows the detailed flow of the signal processing together with a corresponding description.

Figure 4 shows the proposed signal processing flow of the micro-Doppler-based motion feature extraction part. The process has four steps, as shown below.

First, in the pre-filtering step, from the I and Q signals, the DC and high-frequency components are removed using a DC filter and an LPF (Low Pass Filter), respectively. In the paper, the DC filter is implemented by subtracting the average value from the raw signal. We also design the LPF on the 64th order with a cut-off frequency $f_{h,m}$ of 30 Hz.

Second, we conduct the Fourier transform by multiplying the coefficients of a Hamming window with $N_{win,m}$ to suppress the side-lobe of the frequency spectrum and the FFT (Fast Fourier Transform) with $K_{m}$ points for conversion to the frequency domain.

For a detailed explanation of the Fourier transform step, we illustrate the data processing timing flow of the original raw signal in Figure 5.

We employ the STFT (Short Time Fourier Transform) technique based on a sliding window. Because the required Doppler frequency resolution used to separate motion and vital signs is set to approximately 0.5 Hz, we select a window size $N_{win,m}$ of 2000, which means that the measurement time has a $t_{win,m}$ value of 2 s. Moreover, because the maximum Doppler frequency of a passenger’s motion is about 10 Hz, we set the sliding step time $t_{step,m}$ to 0.1 s for sliding window, indicating $N_{step,m}$ sample of 100.

Thus, in this paper, a Doppler spectrum with 2048 FFT points is generated from 2000 samples of the original signal. This procedure is repeated continuously with a sliding window in 0.1 s steps, as shown in Figure 5.

Next, in the micro-Doppler generation step, the magnitude of the Doppler frequency spectrum is calculated using the root-square function, and then the background noise is removed using the previously measured noise spectral distribution information. The generated Doppler frequency spectrum is saved into FIFO (First Input First Output) memory, and the micro-Doppler image is composed by each Doppler spectrum such as the right side of Figure 4. Here, FIFO memory can store as many Doppler spectra as the number of sub-frames L.

Finally, in the motion feature extraction step, as shown in Figure 4, we can obtain two feature vectors by analyzing the distribution of the Doppler scattering points over all sub-frames. This procedure is carried out in a Doppler scattering point analyzer, which will be described later and is shown in Section 2.3.

Figure 6 shows the procedure of the Doppler frequency-based vital sign feature extraction part, which is a simple technique. The process has also four steps, as shown below.

The roles of the pre-filtering step and the Fourier transform step are identical to those in the case of Figure 4. However, the LPF is on the 32th order with a cut-off frequency $f_{h,v}$ of 1 Hz. Moreover, the length of the Hamming window and the number of FFT points are expressed by $N_{win,v}$ and $K_{v}$, respectively.

We also employ STFT technique with a sliding window, as shown in Figure 7. Here, because the maximum Doppler frequency of human vital signs is less 1 Hz, the sliding step time $t_{step,v}$ is set to 1 sec, with reference to a $N_{step,v}$ sample of 1000.

Moreover, for vital sign signals, because the minimum frequency is bounded in the range of 0.1 Hz to 0.5 Hz, the window size $N_{win,v}$ is set to 8000. Thus, we measure the received signal during a $t_{win,v}$ time of 8 s in order to confirm the presence of a breathing signal. As a result, the FFT point is set to 8192.

Next, in the Doppler frequency selection step, the absolute values are calculated as the Doppler spectrum, and rectangular windowing is applied in the frequency domain in order to consider only values below the $f_{h,v}$ frequency, such as the right side of Figure 6.

Finally, in vital sign feature extraction step, as shown in Figure 6, we can set a threshold of scattering points with a magnitude greater than the noise, and we determine vital signs based on survival values. The details pertaining to this are described later and are shown in Figure 8.

As explained in Figure 5 and Figure 7, while the Doppler spectra for motion and vital signs are generated every 0.1 s and 1 s, respectively, according to the step time of the sliding window. That is, while one Doppler spectrum for vital sign detection is generated, ten Doppler spectra for motion detection are completed. Thus, in order to synchronize the two parts and effectively recognize human motion, we generate a micro-Doppler image using ten sub-frame spectra.

### 2.3. Proposed Feature Vector Extraction Scheme

As explained above, because the maximum Doppler frequency of human vital signs is less 1 Hz, in our design, the update time of vital sign detection is 1 sec. In a previous study [13], vital signs were analyzed every one second. Thus, in this paper, we generate a micro-Doppler image for human motion detection every second to synchronize the update time with the vital sign extraction point.

Figure 8 shows the timing diagram used to synchronize the micro-Doppler image for motion detection and the Doppler spectrum for vital sign detection. In this case, the data collection time is initially set to 8 s to generate the first frame’s Doppler spectrum for vital sign detection.

For a detailed explanation of the motion feature extraction process, we define the data stored in FIFO as $\left\{X_{m}\left(i,j\right),\;i=1\sim K_{m},\;j=1\sim L\right\}$, where $K_{m}$ is the Doppler-bin size and L is the number of sub-frames. In this paper, because we set $K_{m}$ to 2048 and L to 10, the size of the micro-Doppler image in one frame is 2048 by 10.

To extract the first and second feature vectors, we first count the number of scattering points stored in FIFO memory every sub-frame, which is expressed as Equation (1). From Y(j), we can obtain the corresponding count value in the *j*th sub-frame.
(1)$$Y\left(j\right)=\mathrm{Count}\left\{X_{m}\left(i,j\right)>0\right\}\;\mathrm{for}\;i=1\sim K_{m}$$

While the vital sign signals of a non-moving passenger or the Doppler signal of another moving object have a narrow distribution of the frequency spectrum, a wide Doppler spectrum can appear when a human moves on the seat of the vehicle. In this paper, to express these characteristics, we define the ‘extended degree of scattering points’ as the first feature, $x_{1}$,. This is expressed by Equation (2), which indicates the average number of scattering points of all sub-frames.

In addition, while the Doppler signals from the breathing of a still human are mostly maintained over time, the passenger’s movements may not be continuous. Thus, the distribution of the Doppler scattering points can vary over the sub-frames. Thus, in this paper, we define these characteristics as the ‘different degree of scattering points’, i.e., as the second feature $x_{2}$. Equation (3) is used to solve $x_{2}$, which is obtained by averaging the differences between the numbers of scattering points of two consecutive sub-frames.

These two feature vectors make it possible to determine whether or not a moving object on a seat in a vehicle is human.
(2)$$x_{1}=\left\{\sum_{j=1}^{L}Y\left(j\right)\right\}/L$$
(3)$$x_{2}=\left\{\sum_{j=1}^{L-1}\left|Y\left(j+1\right)-Y\left(j\right)\right|\right\}/\left(L-1\right)$$

Next, we define the Doppler frequency spectrum as $\left\{X_{v}\left(i\right),\;i=1\sim K_{v}\right\}$ to process the results of the Doppler frequency-based vital sign feature extraction, where $K_{v}$ is the Doppler-bin size. That is, we count the number of scattering points with values greater than reference value for noise thresholding. The reference value can be obtained as the measured noise signal in blank space.
(4)$$Z=\mathrm{Count}\left\{X_{v}\left(i\right)>\mathrm{reference}\;\mathrm{value}\right\}\;\mathrm{for}\;i=1\sim K_{v}$$

Using the results of Equation (4), we define $x_{3}$ as the ‘presence of vital signs’, and the third feature is expressed in Equation (5). When using the feature vector $x_{3}$, we can determine whether or not a non-moving or slightly moving object is a human.
(5)$$x_{3}=\{\begin{array}{l} \mathrm{if}\;Z>0,\;\mathrm{then}\;\mathrm{logic}\;‘1’\\ \mathrm{else}\;\mathrm{logic}\;‘0’\; \end{array}$$

The three extracted feature vectors are fed into the machine learning engine for learning and testing. In this paper, we employ a BDT (Binary Decision Tree) as the machine learning method.

The BDT is a popular and simple machine learning algorithm based on a sequential decision process because a feature is evaluated as one of two branches, which is selected starting from the root of the tree. Thus, we can easily implement the BDT in an embedded system for machine learning based on the three features proposed in this paper, using only the “if–else” syntax in real time [15].

## 3. Measurement Results

### 3.1. Radar Sensor and Measurement Environment

To verify the proposed algorithm, we established a test-bed in the DGIST lab, as shown in Figure 9. The test-bed is composed of a Doppler radar FEM (Front-end Module) with antennas, a DAQ (Data Acquisition) module, a power supply, and a PC.

The radar antennas are mounted to face the vehicle seats, and the transmitted and received ports are connected to the FEM using SMA (Sub Miniature type-A) cables. The received baseband signals are logged by the DAQ module, the data are sent to the PC through a USB (Universal Serial Bus) interface. To control DAQ module and obtain the data in real time, we developed DAQ software using NI’s LabVIEW tool on the PC.

For the purposes of this paper, the FEM and antennas were manufactured by Yeungnam University. In the FEM, a VCO (Voltage Controlled Oscillator) is added, which is different from the board in an earlier circuit version [12]. A photo is shown in Figure 10.

The detailed specifications are presented in Table 1. The center frequency is 2.45 GHz and the FOV (Field of View) of the antenna is 80 degrees. In NI’s DAQ module used here, we set the sampling rate to 1 kHz and the input dynamic range to −5~5 V through the LabVIEW tool.

In addition, the LPF and FFT parameters for the signal processing described above are also shown in Table 1. For the 2.4 GHz Doppler radar system, because the Doppler frequencies of passenger motion and breathing do not exceed 10 Hz and 1 Hz, we select the cut-off frequencies shown in Table 1. The FFT points are also selected such that they support the resolution of the Doppler frequency.

### 3.2. Measurement Scenarios

To verify the proposed human recognition scheme, we designed eight cases as scenarios to be carried out on the test-bed. A photo of each case is presented in Figure 11. In this paper, we measure the radar signal for 60 s in every case. That is, in each case, measurements were conducted for 60 frames. However, in each case, the number of the extracted feature vectors is 52, because of the initial data collection time of 8 s, as described in Section 2.3.

In this paper, we only consider humans as living creatures. That is, we do not discuss other living creatures such as companion animals, because the recognition of the passenger as a human is most important. Moreover, we considered the seats in the vehicle, a box on a seat, and the vehicle body itself as inanimate objects. A detailed description of the scenario is given below.

Case #1: Empty seat without any objects.Case #2: A still human on a seat; for example, a passenger who is sleeping without any motion on the seat.Case #3: A human moving his neck slightly on a seat; for example, a passenger who is dozing with neck movement.Case #4: A human making slight motion on a seat, such as a passenger who is looking around, talking with hand gestures, or has light body movements.Case #5: Humans making more than slight motions on the car seat, such as a passenger who is listening to music with the head or body moving slightly in a wavy motion, shaking leg, keeps moving his body, or has additional body movements.Case #6: Still box on a seat.Case #7: Slightly wobbly box on a seat, such as when a box on the seat is lightly shaken by vehicle vibration; this scenario is virtually simulated by connecting a string to the box.Case #8: Vibrating vehicle such as when the vehicle itself vibrates while driving; this scenario is virtually simulated by actually shaking the test bed.

For the measurements of cases #2–#5, we conducted the experiments with three males. The characteristics of these participants are given below. The photos in Figure 11 show human #2. In the experimental results in Section 3.3 and Section 3.4, we present the results from human #2.

Human #1 is 27 years old, 183 cm tall, and weighs 78 kg.Human #2 is 37 years old, 176 cm tall, and weighs 80 kg.Human #3 is 47 years old, 179 cm tall, and weighs 90 kg.

The CW Doppler radar operates at the 2.45 GHz ISM band, which is freely used for the purpose of industrial, science, and medical applications. The transmitted power of the radar is less than 5 dBm, which meets the regulation for human bodies. All the subjects joining in the experiment agreed with protocol, procedure, and process in the measurement.

### 3.3. Pre-Processing Results

Figure 12 shows the received raw signal for a set time of eight seconds for all cases, where the x-axis and y-axis correspondingly indicate the time (s) and amplitude (voltage level).

In Figure 11a,f, only noise is found, since there are no moving objects (cases #1 and #6). While a breathing signal appears as a sine wave for the human without motion in Figure 11b, we find that the breathing and motion signals are mixed due to the slight movement of the neck (case #3) in Figure 11c.

In Figure 11d,e, because more human motion exists, aperiodic signals are displayed with a higher frequency than the respiration of a human.

Finally, Figure 11g,h show periodic vibration signals that appear due to the shaking of the box and vehicle (cases #7 and #8).

Figure 13 shows a micro-Doppler image generated from the micro-Doppler-based motion feature extraction part described in Figure 4. Figure 12a–h are the signal processing results of Figure 12, showing the results for the eight aforementioned cases. Here, the x-axis is the time (s) and the y-axis is the frequency (Hz).

In Figure 12a,f, no scattering points higher than the background noise are seen because there is no moving object.

In Figure 12b,c, because the human has no movement or only slight motion, mostly breathing signals are extracted. In this case, Doppler spectra with a narrow shape appear almost continuously over time.

Interestingly, in Figure 12g,h, sharp patterns appear, similar to those in Figure 12b,c. This occurs because other objects do not have multiple scattering points.

On the other hand, in Figure 12d,e, a wide distribution of scattering points is found due to various components of human movement. In addition, it can be seen that the distribution of the scattering points varies over time.

Based on these micro-Doppler images, we can extract two feature vectors $x_{1}$ and $x_{2}$ as the ‘different degree of scattering points’ and the ‘extended degree of scattering points’ through processing, as shown in Figure 8.

Figure 14 shows the Doppler spectra of the Doppler frequency-based vital sign feature extraction part shown in Figure 6. These results are also measured based on the signals in Figure 12 for all eight scenarios. Here, the x-axis is frequency (Hz) and the y-axis denotes the magnitude. In addition, the red-dotted boxes indicate the area of the frequency below 1 Hz, occupied by the breathing signal.

In Figure 14a,f, no dominant spectrum is found due to the absence of motion or vital sign. In Figure 14b,c, the Doppler spectrum is sharp in the red box due to the breathing signal. However, the spectrum of Figure 14c is slightly widened compared to that in Figure 14b due to the motion of the neck.

In Figure 14g,h, sharp-type Doppler spectra are also found, but most of them are located outside of 1 Hz.

On the other hand, in Figure 14d,e, we find that the Doppler spectra are spread across the red box. Occasionally, maximum peaks can be found at less than 1 Hz according to the Doppler volume of human movement.

Based on these Doppler spectra, we can determine whether or not vital signals exist, and obtain the third feature vector $x_{3}$. In this paper, a simple algorithm that can extract the breathing signal is employed. If we use a very fine algorithm, we can extract vital signs with a high detection probability.

Figure 15 shows two motion features extracted from micro-Doppler images of Figure 13 for eight cases. Figure 15a,b indicate the first feature $x_{1}$ as the ‘extended degree of scattering points’ and $x_{2}$ as the “different degree of scattering point”. Here, the x-axis is the time (sec) and the y-axis is the number of the detected scatters. The results for cases #1 to #8 are correspondingly represented by the green-solid, red-solid, red-dotted, blue-solid, blue-dotted, green-dotted, black-solid, and black-dotted lines.

Figure 15a shows that the number of scatters in cases #2 and #3 is mostly lower compared to cases #4 and #5 due to the different levels of human movement. That is, for a still or slightly moving human, only a few scattering points are reflected.

In cases #7 and #8, we can find similar patterns to those in cases #2 and #3. In cases of inanimate objects, the Doppler spectra are narrow, because multiple components do not exist.

Finally, when there is no motion component, the number of extracted features is 0, such as in cases #1 and #6.

In Figure 15b, when the passenger is moving on a seat for cases #4 and #5, we find that the distribution of the Doppler spectrum varies more than those of a still human (case #2), a slightly moving human (case #3), and other objects (cases #7 and #8).

Figure 16 shows the extracted Doppler frequency spectra for the third vital sign feature in Figure 14. That is, Figure 16a,h indicate the presence or absence of vital sign. For all cases, the x-axis and the y-axis denote the time (s) and logic value (true or false), respectively.

As shown in Figure 16a,f, no signal is detected in the cases without motion

In Figure 16b,c, vital signals are recognized at all times, despite the fact that the passenger moves his neck only slightly. However, from the results in Figure 16d,e, we find that breathing signs may or may not be extracted depending on the movement level of the human.

Finally, for inanimate objects, we can find that vital signals are mostly not detected, as shown in Figure 16g,h. However, in the results, false detections occasionally occur due to noise. This problem will be resolved by employing a fine breathing detection algorithm in the future.

### 3.4. Proposed Feature-Based Human Recognition Results

Figure 17 presents the three-dimensional distributions of the features extracted from the micro-Doppler image and the vital sign frequency spectrum.

In cases #1 and #6, without any moving object, the three features are positioned at zero, as shown by the purple boxes.

We can distinguish between a human with no or little motion (cases #2 or #3) and an inanimate object with movement (cases #7 and #8) using only the vital sign feature $x_{3}$. Here, cases #2 and #3 are displayed with red-star marks, and the blue-cross marks are used to present cases #7 and #8. However, as mentioned above, even in cases #7 and #8, a few incorrect results appear, as if vital signals are detected due to noise.

As shown by the green circle marks in cases #4 and #5, because the $x_{3}$ values for the passenger with movement are distributed between 1 and 0, it is impossible to determine whether the detected object is human or not if using only $x_{3}$. However, in these two cases, the first and second features extracted from the micro-Doppler image are positioned in an area far from the origin, while the results for the inanimate object appear around the origin. Thus, when using $x_{1}$ and $x_{2}$, we can distinguish between a human with motion and other objects with movement.

In this paper, we use three feature vectors to train and test the process using machine learning with the BDT, as shown in Figure 18. The procedure we used for all programming for machine learning and verification is described below. Here, we coded all procedures using the Matlab library.

We labeled the features of the actual human as ‘1’ for 208 frames and other cases as ‘0′ for 208 frames, respectively. Here, one frame was measured every one second, as shown in Figure 8.We then randomly separated the three features for the human cases and the others, with 80% for the training data set and 20% for the test set. This means that 80% of the total 208 frames with label ‘1’ were used for the training set, with the remaining 20% being allocated to the test set. Moreover, among the feature vectors of 208 frame times with the label ‘0’, the data of 166 s and 42 s were used for training and test, respectively.We optimized the machine learning engine of the BDT library via a 30-trial loop with the training data set.We input the test set into the optimized machine learning engine.We repeated the three-step procedure described above ten times, while also dividing the data set, training, and testing steps.We checked the performance by averaging the results of the ten aforementioned trials.

In the typical methods [7,12], vital sign monitoring of breathing in vehicle applications is used. That is, the sampled radar echo signal is analyzed for the presence of periodic breathing while separating vital signs form background noise. Thus, previous works considered only the scenario of a human being asleep in a vehicle. In this paper, we define instances that use only vital sign signals as the typical method.

In Table 1, the human recognition performances outcomes are presented for the typical method and with the proposed algorithm. In the typical algorithm, only the vital signals are used in cases for a human with no motion and with slight motion. However, in the proposed algorithm, we use not only the characteristics of the Doppler scattering points, but also the breathing signals.

We present two performance metrics: the classification accuracy (%) and the classification error rate (%).

The classification accuracy indicates whether or not an actual human has been accurately classified as a human. On the other hand, the classification error rate represents the rate at which an inanimate object is mistakenly determined to be a human.

While the classification accuracy of the typical algorithm using only vital signs is approximately 70%, the classification accuracy of the proposed method is improved to 98.6%, as shown in Table 2. That is, the performance is enhanced by nearly 28%.

Regarding the classification error rate, the performance of the proposed method is decreased by 0.5% compared to the typical method. In this paper, because we employ a very simple algorithm to detect vital signs, the noise of an inanimate object can be occasionally incorrectly recognized as vital signs. If the vital sign detection algorithm is advanced in the future, this problem will be resolved.

In this paper, we conducted this experiment with three people, and similar performance outcomes were obtained. This occurred because the measuring distance is very close, at about 1 m, and the shapes of the human bodies of the participants are similar.

## 4. Conclusions

In this paper, we defined three new features and proposed a human recognition scheme based on machine learning using a CW radar sensor. To do this, we initially measured the ‘extended degree of scattering points’ from micro-Doppler images, after which we calculated the mean of the Doppler reflection points over sub-frames. Second, in order to extract the ‘different degree of scattering points’, we calculated the mean of the difference in the Doppler reflection points between two successive sub-frames.

While the two feature vectors described above were meant to recognize a human’s motion, the last feature vector is for human vital sign recognition. Hence, we defined the ‘presence of vital signs’ as extracted from the Doppler frequency spectrum as the breathing signal of a human.

To verify the performance of the proposed algorithm, we built a test-bed similar to the interior of an actual vehicle and defined eight cases, consisting of non-moving objects, a still or moving human, and inanimate moving object scenarios. For these cases, we used a commercial real-time DAQ module and a 2.45 GHz CW radar front-end module with antennas developed by Yeungnam University. Then, in order to extract three features from the received radar signals, we obtained the raw data using the test-bed.

The extracted features were used as input data for a BDT as machine learning engine, and we verified the proposed algorithm through randomly repeated verification trials.

The results with the typical method using only vital signs show that the classification accuracies for a human were 70.7%. However, with the proposed human recognition scheme using motion and vital sign features, the classification accuracy was found to be 98.6%. That is, compared to the typical method, the performance of the proposed method is improved by approximately about 28%.

Moreover, because the proposed algorithm has very low complexity, we can implement a passenger detection radar system with a simple structure.

In the future, using radar sensors with multiple receiving antennas, we will conduct research to determine the presence and status of occupants for each seat. In the addition, we plan to employ a new vital sign detection algorithm to improve the classification error rate for humans. We will also verify the proposed algorithm together with the various types of human forms. Moreover, we will install the test-bed in an actual vehicle in order to verify the proposed recognition scheme more practically.

## Figures and Tables

**Figure 1 sensors-20-06202-f001:**
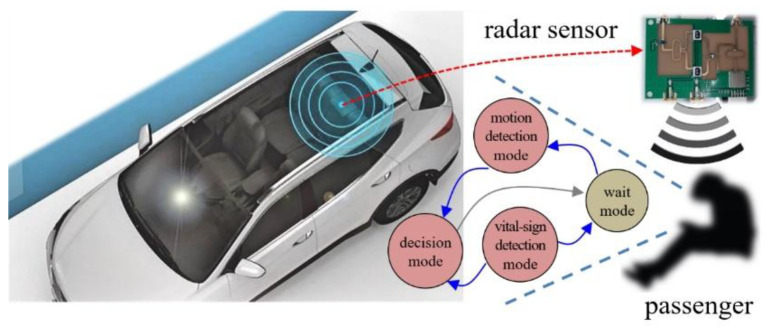
Human recognition concept for in-vehicle application. Here, the car photo on the left, also used in an earlier work [2], was modified somewhat.

**Figure 2 sensors-20-06202-f002:**
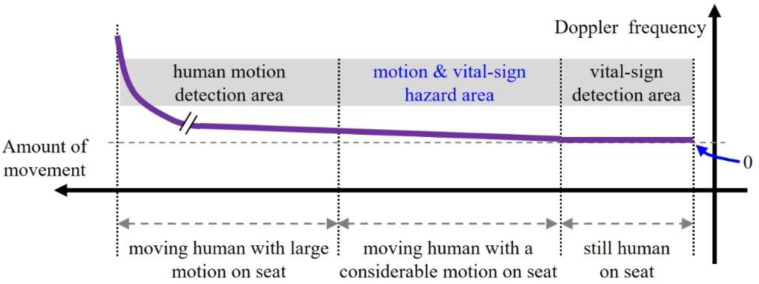
Information detected from a radar sensor according to amount of movement by a human on a seat in the vehicle.

**Figure 3 sensors-20-06202-f003:**
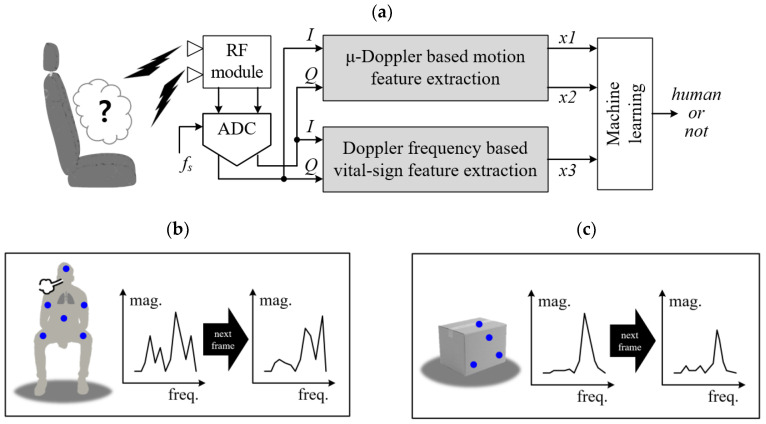
Proposed human recognition scheme together with micro-Doppler and vital signals using machine learning for a Doppler radar sensor: (**a**) top block diagram of the proposed algorithm, (**b**) example of Doppler scattering points of a human, (**c**) example of Doppler scattering points of a non-human and a non-human object.

**Figure 4 sensors-20-06202-f004:**
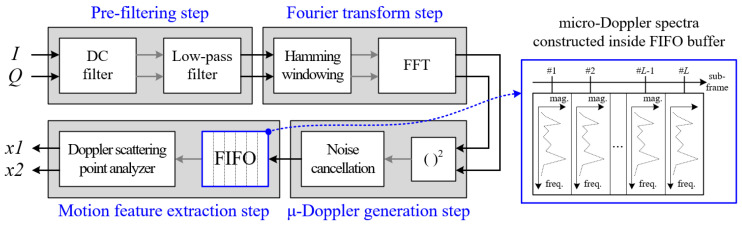
Detail algorithm steps for micro-Doppler-based motion feature extraction.

**Figure 5 sensors-20-06202-f005:**
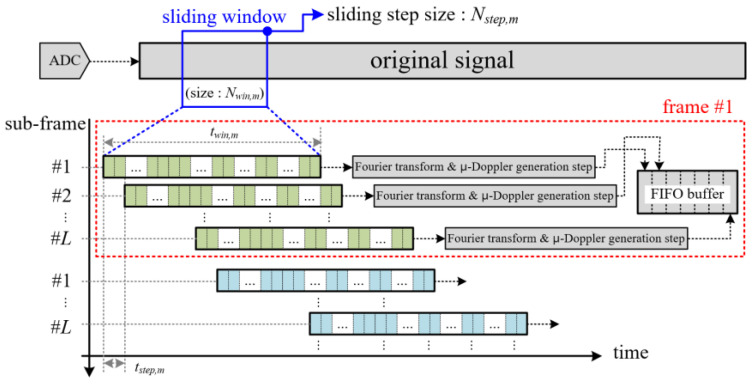
Concept of sliding window-based micro-Doppler generation for motion feature extraction.

**Figure 6 sensors-20-06202-f006:**
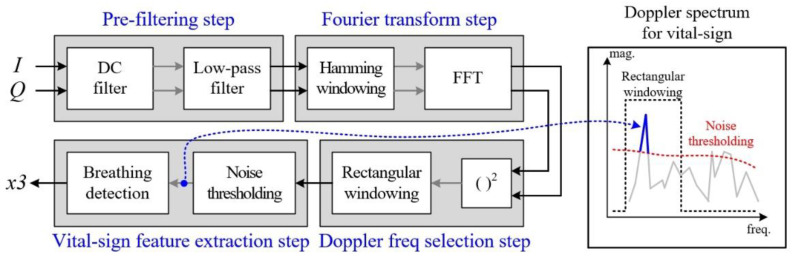
Detailed algorithm steps for Doppler frequency-based vital sign feature extraction.

**Figure 7 sensors-20-06202-f007:**
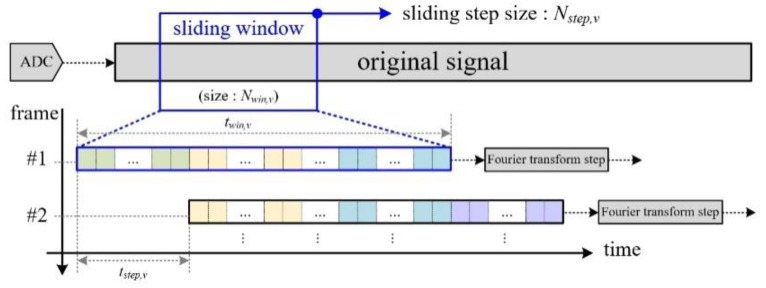
Concept of sliding window-based Doppler frequency extraction for vital sign feature extraction.

**Figure 8 sensors-20-06202-f008:**
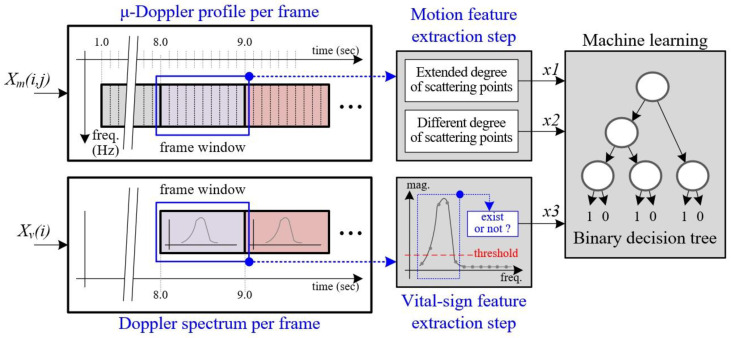
Process of extracting both motion and vital sign features from the micro-Doppler and Doppler spectra with the same time interval.

**Figure 9 sensors-20-06202-f009:**
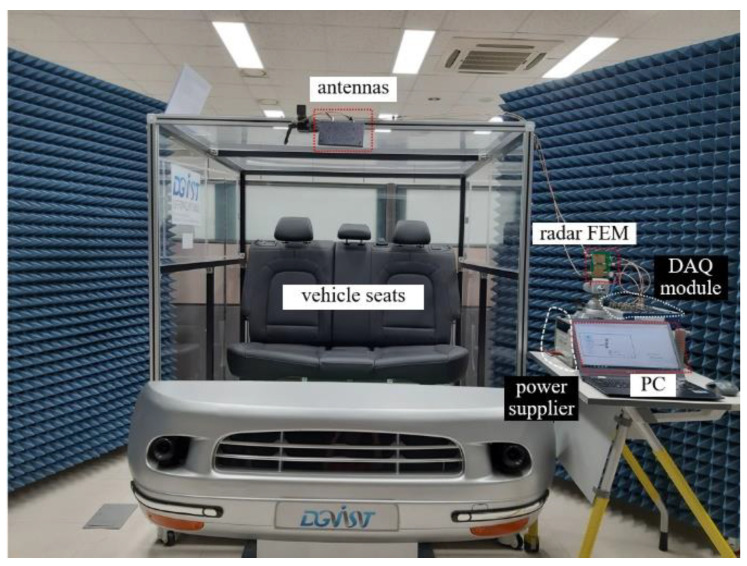
Photo of the test-bed built in the DGIST lab.

**Figure 10 sensors-20-06202-f010:**
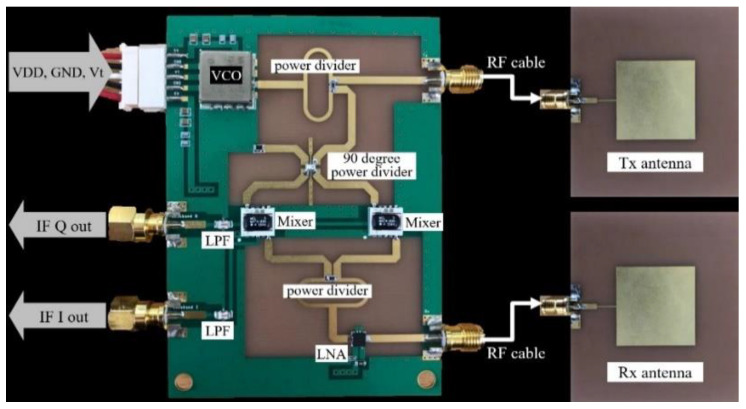
Photo of the 2.45 GHz CW radar prototype.

**Figure 11 sensors-20-06202-f011:**
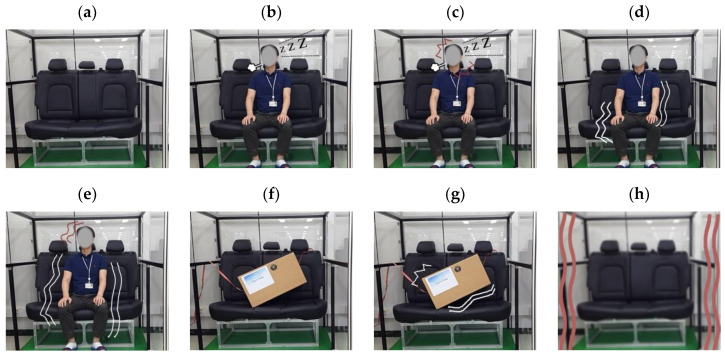
(**a**–**h**) Test scenarios represented as eight cases in a vehicle.

**Figure 12 sensors-20-06202-f012:**
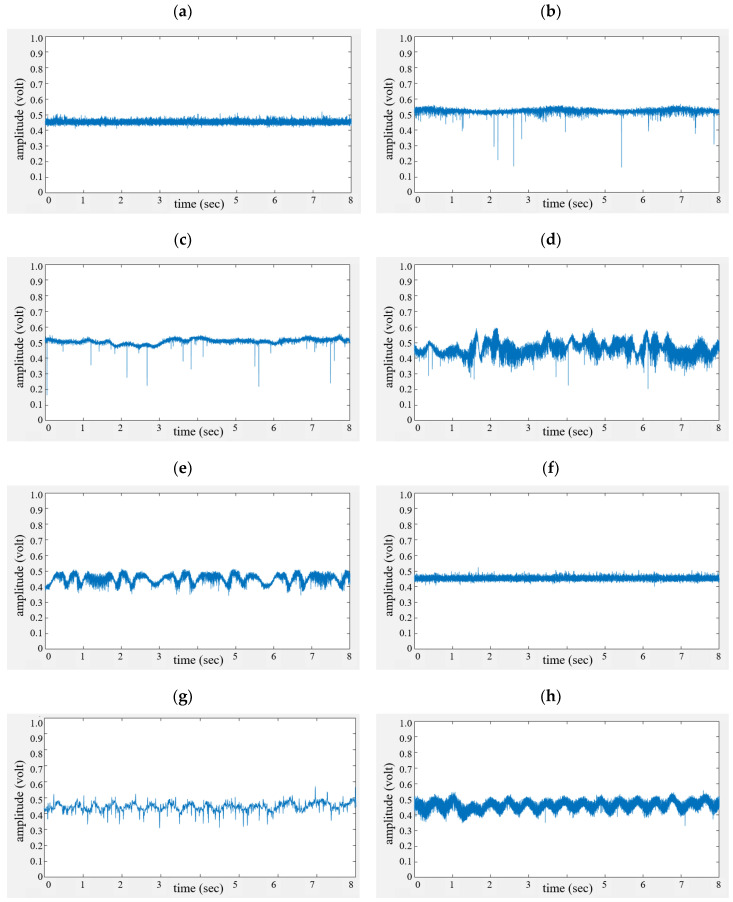
Received signals in the time domain; (**a**–**h**) indicate cases #1~#8.

**Figure 13 sensors-20-06202-f013:**
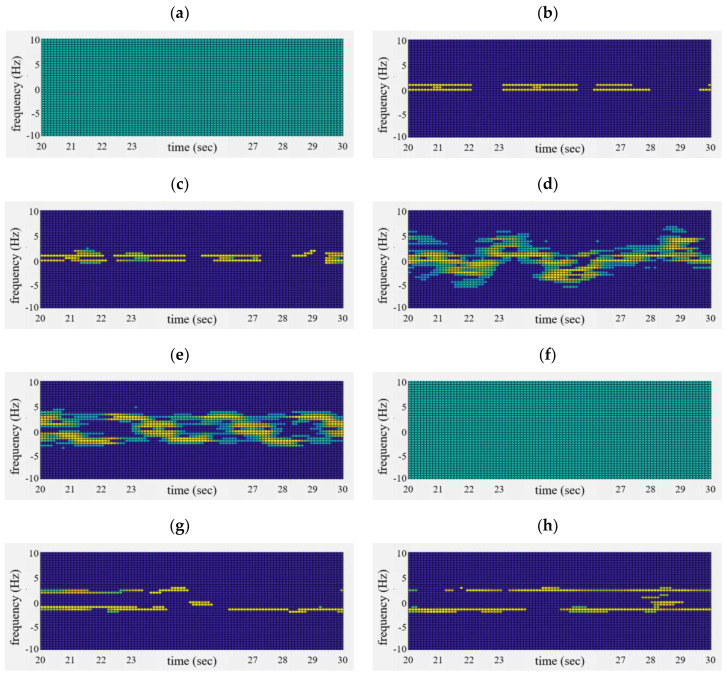
Extracted micro-Doppler image of μ-Doppler-based motion feature extraction part; (**a**–**h**) correspondingly indicate cases #1 to #8.

**Figure 14 sensors-20-06202-f014:**
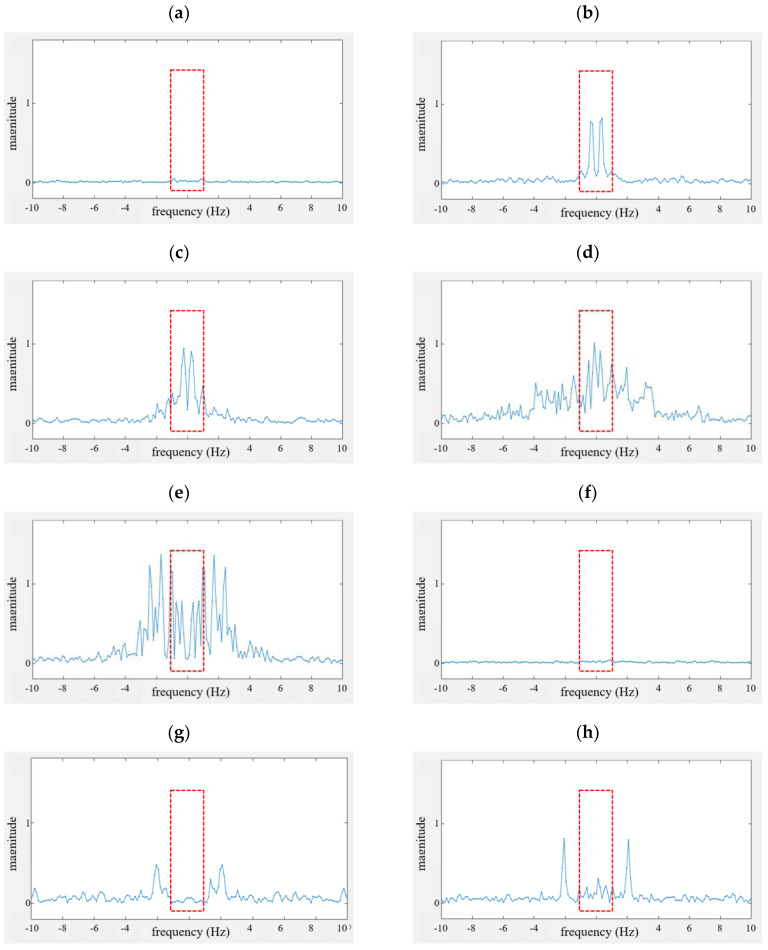
Doppler spectrum extracted from the Doppler frequency-based vital sign feature extraction part; (**a**–**h**) indicate cases #1 to #8.

**Figure 15 sensors-20-06202-f015:**
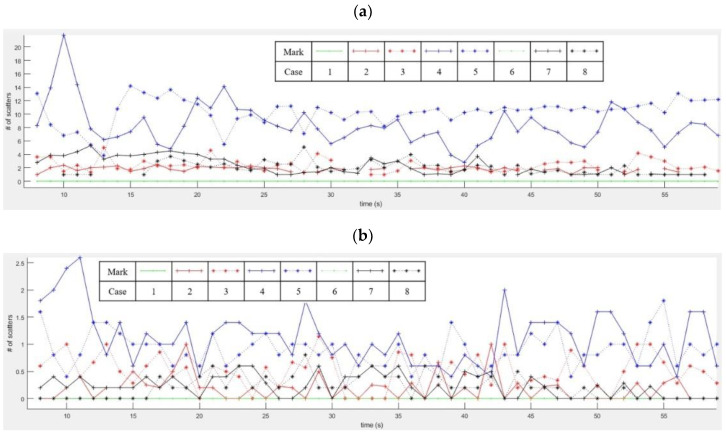
Motion feature extraction results for eight cases: (**a**) results of the extended degree of scattering points and (**b**) the results of the different degree of scattering point.

**Figure 16 sensors-20-06202-f016:**
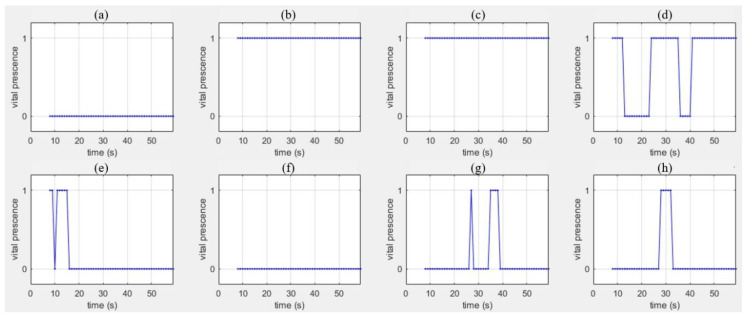
Vital sign feature extraction results; (**a**–**h**) indicate cases #1 to #8.

**Figure 17 sensors-20-06202-f017:**
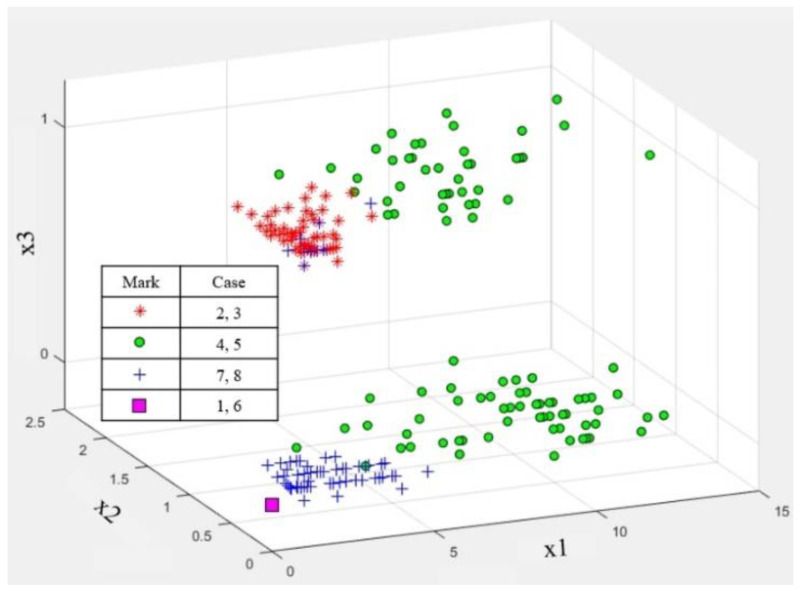
Three-dimensional distribution of three features extracted from the proposed algorithm scheme for a human and other objects.

**Figure 18 sensors-20-06202-f018:**
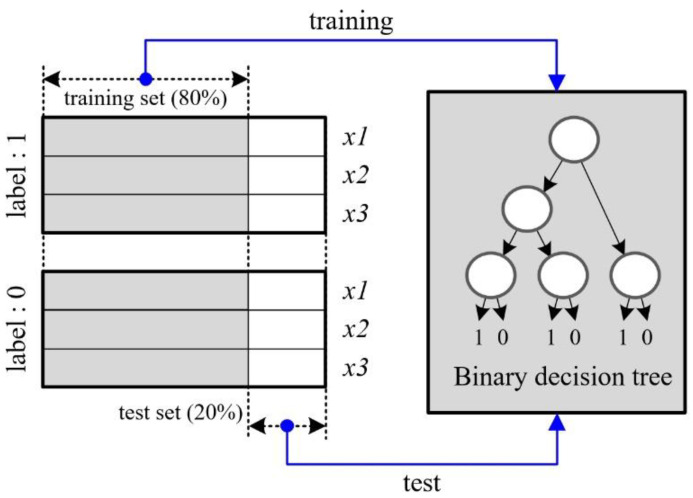
Structure of features comprising the training data set and the test set for machine learning.

**Table 1 sensors-20-06202-t001:** Parameters of the radar system used in this paper.

Parts	Specifications		Units	Symbols	Values
FEM/antennas	Center frequency		GHz	f_c_	2.45
	Field of view		Degree	-	80
DAQ module	ADC frequency		MHz	f_s_	1
	Dynamic range		V	-	−5~5
Signal processing	Motion detection	Cut-off frequency	Hz	f_h,m_	10
		FFT point	point	K_m_	2048
	Vital sign detection	Cut-off frequency	Hz	f_h,v_	1
		FFT point	point	K_v_	8192

**Table 2 sensors-20-06202-t002:** Classification performance for human recognition.

	Performance	Classification Accuracy (%)	Classification Error Rate (%)
Metrics Algorithm			
typical method		70.7	4.8
proposed method		98.6	5.3

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
