# Peer review of "Machine Learning-Based Human Recognition Scheme Using a Doppler Radar Sensor for In-Vehicle Applications"

_sensors, 2020, doi:10.3390/s20216202_

Round 1

Reviewer 1 Report

The authors present a scheme that uses Doppler radar sensors and machine learning in order to detect the presence of humans in vehicles. The paper is overall in good shape, however, there are a few issues that would make the manuscript a sounder one. Those are as follows:

  1. The abstract needs rewriting. It really does not reflect nor represent the rest of the paper, if anything it misleads the reader. For example, lines 18-20 make no sense: what does it mean to use binary decision tree FOR machine learning? Also, in abstract, the authors speak about newly defined features and then about the verification of a proposed algorithm, about which we haven't heard to that point. Also, the abstract fails to answer the crucial question: what is that you are trying to detect and learn?
  2. The motivation is clear and well-founded, however, I would like to point out that perhaps the proposed work is not just for human but in general for living creatures. Would that change the research ?
  3. Figures showing the Doppler component and echo levels in cases of living and non-living objects (as shown in the experiments section) would add value to the paper if presented also early in the paper to illustrate and solidify the motivation.
  4. Is there a point in including both figures 5 and 7? They seem exactly the same.
  5. In the experiments section, what is the case of a living creature that is present but not breathing? Does that appear in the signals as a non-living object? Would it make sense to distinguish in order to detect an emergency?
  6. I strongly recommend to use standard classification performance evaluation using precision/recall/accuracy. 
  7. There is no information in the experiments set-up with regards to the number and types of objects recorded/monitored. How many humans? How many different humans? What are the 80% (and 20%) training points in terms of data? 
  8. There is no related works section. Even if that is the first work using Doppler sensors for this particular problem, as mentioned in the introduction, other sensors have been extensively used in literautre for detecting presence and occupancy in vehicles. It is paramount to discuss those more closely, but also to compare the proposed work with some of them. 
  9. You compare your proposed algorithm with "the typical method". Is there some citation for this? 

Author Response

We appreciate the reviewers for their great valuable review of our paper.

We also would like to acknowledge the constructive comments made by the reviewers.

We have revised the manuscript by incorporating these suggestions.
Moreover, all additions and corrections are indicated in the revised paper.

Reply of each comment is attached with the additional file.

Reviewer 2 Report

The paper discuss a model of intelligence trained to recognize the load position in the car. Idea of the system is interesting with practical experimental verification in laboratory conditions. Some aspects of the paper should be revised:

  1. The Doppler models data acquisition is not clear enough. From fig. 3 we see that there are some measure points on passengers body. Are these points sensors or your system is measuring body in these points? How do you measure values?
  2. Have you tried any other transforms in your model. Sometimes wavelet model gives even better results than classic approaches. Have you done any research in this field?
  3. Some extension to introduction is important: Body pose prediction based on motion sensor data and Recurrent Neural Network, Intelligent Internet-of-Things system for smart home optimal convection.
  4. From fig. 8 we can see that frame measure window is 1 sec. Is this an optimal size? Have you tried other configurations?
  5. From fig. 8 we see that final result is based on machine learning approach. What system was used, which methods were used in training, how did you configure your model? There is no presentation of your applied machine learning, please refer to this and discuss it better.
  6. There are no comparisons to other models so we don’t know if your machine learning model is better or not. Explain this in details.

Author Response

(The authors gave the same response as above.)

Reviewer 3 Report

1. The evaluation part is weak and the quality can be improved. The discussion and evaluation part only considers several different parameter settings for the authors’ solution and compares this work to the typical scheme. How the proposed protocol is strengthening the state of the arts? 2. This paper should carefully review the class of these related and previous works (e.g., ref. [6-7], [9-10], [13-15],) and clearly give the differences and major improvements of the proposed mechanism. Authors should provide more convincing reasons to readers. Without this work, the reviewer cannot clearly identify the contribution of this paper.

Author Response

(The authors gave the same response as above.)

Reviewer 4 Report

The paper gives an interesting research about the human recognition in vehicle using CW radar. The authors have explored several new features and the results are very well. However, the paper seems more like a report rather than an academic article. The writing should be greatly improved. For example, the paper should be divided into more sub-section. More subtitles should be added.

The current paper records what authors have done clearly. But the problems to be solved, modelling, methodology are not easy to find. The paper organization should also be improved.

Some other comments:

Figure 4-5 and Figure 6-7 are similar and can be merged.

P6, Line 209: Why is the window size 8000 for the 1Hz Doppler frequency resolution but the window size 2000 for the 0.5Hz Doppler frequency resolution in the motion processing?

Author Response

(The authors gave the same response as above.)

Round 2

Reviewer 1 Report

We would like to thank the authors for taking the time and the effort to revise their manuscript as per our comments. The difference is obvious and the new version of the paper is much better than the previous one. However, there is still room for improvement, for example, in the experimental set up, it is still not sufficiently explained what is 80% and 20% in terms of training/testing data. For example, if my experiment involves 500 hours worth of running, then 80% of my data used for training would be 400 hours of these and the rest would be my test set. What is the data in your case?

Author Response

We appreciate the reviewers and the Assistant Editor for their great valuable review of our paper.

We also would like to acknowledge the constructive comments made by the reviewers.

We have revised the manuscript by incorporating these suggestions. Moreover, all additions and corrections are indicated in the revised paper.

Reviewer 3 Report

The authors have satisfactorily addressed the main concerns.

Author Response

We appreciate the reviewers and the Assistant Editor for their great valuable review of our paper.

Reviewer 4 Report

Authors have replied to the comments well. The current paper is good for readers' understanding. It can be published in the current form.

Author Response

(The authors gave the same response as above.)
